# Chronic diseases: An added burden to income and expenses of chronically-ill people in Sri Lanka

Ruwan Jayathilaka *, Sheron Joachim, Venuri Mallikarachchi, Nishali Perera, Dhanushika Ranawaka

Department of Business Management, SLIIT Business School, Sri Lanka Institute of Information Technology, Malabe, Sri Lanka

\* ruwan.j@sliit.lk

## Abstract

In the global context, health and the quality of life of people are adversely affected by either one or more types of chronic diseases. This paper investigates the differences in the level of income and expenditure between chronically-ill people and non-chronic population. Data were gathered from a national level survey conducted namely, the Household Income and Expenditure Survey (HIES) by the Department of Census and Statistics (DCS) of Sri Lanka. These data were statistically analysed with one-way and two-way ANOVA, to identify the factors that cause the differences among different groups. For the first time, this study makes an attempt using survey data, to examine the differences in the level of income and expenditure among chronically-ill people in Sri Lanka. Accordingly, the study discovered that married females who do not engage in any type of economic activity (being unemployed due to the disability associated with the respective chronic illness), in the age category of 40–65, having an educational level of tertiary education or below and living in the urban sector have a higher likelihood of suffering from chronic diseases. If workforce population is compelled to lose jobs, it can lead to income insecurity and impair their quality of lives. Under above findings, it is reasonable to assume that most health care expenses are out of pocket. Furthermore, the study infers that chronic illnesses have a statistically proven significant differences towards the income and expenditure level. This has caused due to the interaction of demographic and socio-economic characteristics associated with chronic illnesses. Considering private-public sector partnerships that enable affordable access to health care services for all as well as implementation of commercial insurance and community-based mutual services that help ease burden to the public, are vital when formulating effective policies and strategies related to the healthcare sector. Sri Lanka is making strong efforts to support its healthcare sector and public, which was affected by the coronavirus (COVID-19) in early 2020. Therefore, findings of this paper will be useful to gain insights on the differences of chronic illnesses towards the income and expenditure of chronically-ill patients in Sri Lanka.

**Data Availability Statement:** The data underlying the results presented in the study are available from Department of Census and Statistics, Sri Lanka. Interested parties can obtain this data file

after visiting the Department of Census and Statistics in Sri Lanka or following the Microdata Dissemination policy in DCS. Eg: http://nada. statistics.gov.lk/index.php/dissemination https:// unstats.un.org/unsd/dnss/docViewer.aspx?docID= 1766 After requesting for the survey data from Department of Census and Statistics in Sri Lanka, they will evaluate the application before providing the access to the data. Moreover, the authors did have have any special access privileges others would not have.

**Funding:** The authors received no specific funding for this work.

**Competing interests:** The authors have declared that no competing interests exist.

## Introduction

Chronic diseases are conditions such as, arthritis, cardiovascular disease, heart attacks, cancers, epilepsy etc., that last for a period of more than one year and which cannot be fully cured by medication. Chronic diseases, which are also known as non-communicable diseases (NCDs), are rapidly escalating the patient toll. At present, NCDs cause many confrontational effects such as, disability, premature death and high out of pocket healthcare expenses which can lead people into the trap of poverty.

It is estimated that 83% of total deaths in Sri Lanka in 2016 were because of NCDs [1]. Furthermore, the associated costs of these illnesses are immense, thus, create an income and expenditure inequality among the chronically-ill people. For an example, how an executive making Sri Lankan rupees (LKRs.) 500,000 per month vis-à-vis with a teacher who earns SLRs. 50,000 would bear the burden of medical and care expenditure will vary drastically. Significant discrimination is noticeable in the distribution of income among people. Hence, the impact NCDs could make on low and middle-income countries (LMICs) like Sri Lanka remains a challenge. According to the Ministry of Health and Nutrition and Indigenous Medicine [2], it was revealed that the availability of essential laboratory facilities and drugs for various chronic diseases is limited to a high extent. In particular, this finding is valid in terms of main government hospitals which offers health care services to the public for free, specially in countries like Sri Lanka. However, service quality, accessibility, long queues or waiting lists, time convenience are some of the issues faced by patients. This is one of the main reasons for the victims of NCDs to undergo the burden of high out of pocket expenditure, as they tend to seek necessary medication from private hospitals at an extremely high price.

According to the World Health Organization [1], four out of five chronic disease deaths that occur in the world today spur from low and middle-income countries like Sri Lanka. In the year 2015, 400,000 chronic patients were reported from Sri Lanka [2]. As such, this study has been executed as an original in the context of Sri Lanka, based on the Household Income and Expenditure Survey (HIES) 2016 data set. Purpose is to investigate the differences in the level of income and expenditure among this significant toll of chronic patients reported from Sri Lanka, which is high in numbers at present. In addition, this study aims to contribute its findings to policymakers and responsible authorities to devise feasible policies and initiatives. It is expected that such policies can support the public of the country to ease the burden of medical expenditure, which they currently bear with great difficulty.

Globally, chronic diseases have affected health and the quality of life of many citizens, where more than two third of total deaths are caused by a certain type of chronic disease. The issues associated with chronic illnesses are projected to rise rapidly in the coming years, especially in developing countries like Sri Lanka. As such, increasing growth in chronic diseases creates significant barriers to growth and development. The low-income households are at risk the most for developing chronic diseases and premature deaths, as they are more vulnerable for several reasons. These include greater and reduced access to healthcare facilities.

In Sri Lanka, the HIES results reveal that out of 82,961 individuals living in the housing units, a total of 11,798 individuals suffer from chronic diseases. In other words, 14.22% of persons are deemed to be victims of a certain type of chronic illness. Nevertheless, the situation is critical, and it can be noted as to how different types of chronic illnesses can difference on income and expenditure levels.

Currently, the relationship between chronic illnesses differencing income and expenditure is a largely unexplored topic to which less attention has been paid. To date, no attempts have been made to explore this area in the Sri Lankan context. Thus, this study will focus on contributing to the above-mentioned research gap.

The objective of this research is to investigate differences in the level of income and expenditure of people diagnosed with chronic illnesses. As such, this research differs from existing studies to date, and contributes to literature in four ways. Firstly, no prior research study has been conducted with regards to the area under consideration, addressing the local arena. According to information available to researchers, this study will be the first attempt of this kind.

Secondly, chronic illnesses being a severe health condition that persists for a period of one year or more, require households to incur continuous caregiving and medical treatments on behalf of their patients. Such treatments are mandatory as these patients can become severely ill and helpless for a considerable period of time. Consequently, this creates numerous barriers to perform routine activities on a daily basis for both parties, i.e. for the sick person and his family members. At present, an individual is prone to be affected by more than one chronic condition when conditions worsen [1]. Therefore, NCDs have become a major health issue in the 21st century which require the attention of all regulatory bodies of a nation such as the government, healthcare sector and other policymakers. Adverse effects of NCDs on economic well-being of individuals, households and the society, can result in a decline in growth prospects of nations. This is noticeable especially in LMICs such as Sri Lanka which is at greater exposure to risks and limited access to better healthcare facilities for the general public.

Thirdly, according to the health goal 'SDG 3' in line with the Sustainable Development Goal (SDG) profile of Sri Lanka issued by the World Health Organization (WHO) South East Asian Region in 2017, the likelihood to die from NCDs before the age of 70 is 17.7%. This number is expected to rise in the coming years [2]. Even though many research studies have been conducted in several other countries with regards to the selected area, no significant research has been carried out in Sri Lanka using the HIES data set. Hence, at the completion of the study, the findings of this research will provide valuable insights for introduction, planning and implementation of new government policies related to healthcare. It can also assist to eliminate or reduce the probability of occurring NCDs among the general public by spreading awareness among the society. Finally, the findings will be helpful, particularly to the healthcare sector which endeavours to recuperate quality of life in the aftermath of the COVID-19 outbreak in early 2020.

The remaining sections of this paper are organized as follows. Section 2 describes the literature review, while Section 3 presents data and methodology. Section 4 assesses the empirical results and the discussion, whereas Section 5 presents the concluding remarks.

## Literature review

Globally, health and the quality of life of many citizens have been affected by chronic diseases. More than two third of total deaths are caused by a certain type of a chronic disease [3]. Over 41 million people have died from NCDs in 2016 which is estimated to be 72% of the global mortality, out of which, 15 million deaths have occurred between the age of 30–70 years. These type of deaths are mainly reported from cardiovascular diseases (31%), cancers (16%), chronic respiratory diseases (7%), diabetes (3%) and other NCDs (15%) [1]. Another study reveals that, the issues associated with chronic illnesses are projected to escalate in the coming years. Such a scenario can create barriers to economic growth. Additionally, chronic diseases have been recognized as the cause for disability for 68% of people living worldwide and 84% in LMICs [4]. Thus, the increasing prevalence of multiple chronic conditions will result in increasing healthcare utilization and thereby increasing costs.

The effect of growth of various chronic illnesses and disorders befall on the world at large. However, the impact it has especially on LMICs is possibly high due to various reasons. In

other words, a study claims that the increasing global burden of diseases due to all chronic ill-ness conditions may impose a heavy financial burden on households in LMICs [5]. Proving the above fact further, according to Burki, Khan [6], despite high numbers of chronically-ill patients are found among the non-poor, chronic diseases cause severe effects and conse-quences on middle-income and low income earning persons within the country. Therefore, in such a setting, the probability of becoming poor is likely to increase.

In addition, most of the LMICs have a considerably low level of public expenditure, inade-quate health insurance and low coverage of healthcare services compared to well off nations. Insufficiency in public health services and expenditures has caused victims to experience high amounts of out-of-pocket expenditures, compelling them to acquire private healthcare services which are mostly highly expensive and unaffordable [7]. Moreover, it has been proven that medical expenses represent a substantial proportion of economic costs in treating chronic ill-nesses in poor countries. As such, the growth of chronic illnesses will have an adverse impact in the lives of people living in such countries [8]. Mostly, these expenditures consist of medica-tion costs for patients who are in need of regular and on-going treatments, payments for medi-cal and allied health services, purchase of medical devices and other related essential services [9]. Hence, the expenditure on chronic diseases has impacted heavily on households especially in LMICs due to excessive expenses compared to the meagre income they earn.

Studies prove that, when medical treatments turn out to be highly costly to individuals, lat-ter tend to shrink their usage of health services and medications. Considering this, the increas-ing costs are confirmed as a cause for non-adherence by chronically-ill patients, which can ultimately harm the health of patients [9]. This dilemma is not only seen in developing and less developed countries but also in developed countries like Australia. An international survey conducted in Australia reported that 20% of Australians skip medication because of the unbearable costs [10]. The negligence of prescribed medication was found in affluent countries such as Germany, France, New Zealand and in the United Kingdom as well [10]. Hence, peo-ple in developed countries too tend to disregard accessing healthcare services due to penalizing of expenditures (associated with healthcare). This means that the growing financial burden associated with NCDs have an impact on every nation dispersed across the world.

Furthermore, chronic illnesses and disabilities can cause to collapse the economic stability of households by bringing in adverse economic consequences. These includes unemployment, change in status of employment, reduction in employee payment, out-of-pocket medical expenses, home modification expenses, etc. Thus, the impact of chronic illnesses towards the economic growth of a country is high and can be adverse. This is because there is a reduction in labour units as well as capital accumulation [11], These are caused due to treatments being less affordable and other health related setbacks which would restrict the economy moving towards development [12]. Moreover, the heavier the burden, affected people tend to borrow or sell their assets, leading to a long term financial burden ahead due to borrowing and sale of assets [5]. This ultimately drags affected people as well as families into inflaming poverty and trap them in it, therefore not enabling them to be relieved from the so called financial distress. Therefore, high levels of financial stress, medical debt and bankruptcy can be found among people who suffer from chronic disabilities [9], Following this, it again proves the fact that high expenditure which need to be incurred on chronic illnesses have caused an unbearable burden on households in LMICs.

Nevertheless, income level of households is another variable that can lead to chronic induced poverty. A research conducted in 2007 specifies that the prevalence of chronic dis-eases is high among people who live in rural areas than those in urban areas. This is attribut-able to the existence of income inequalities. It also mentions that in Bangladesh, a vast majority of the people who suffer from chronic diseases and associated disabilities fall under

the lowest two wealth quintiles of the society [4]. Another study claims that, Australia being a well-developed nation still struggles with the issue of high medication costs. In this sense, low income earning households without any entitlement for concessions spend 5% to 26% of their total discretionary income on medicines [13].

Furthermore, Lan, Zhou [14] depict that due to the limited income availability for low income earners living in rural areas, such people have a higher tendency to be victims of health payment induced poverty, rather than income earners living in urban areas. The reason is that these people are much closer to the poverty line and thus, more vulnerable to poverty. The findings of another research depict that households are more likely to spend 11% of their total household budget on healthcare and medications, where as 50% of the occupants tend to spend 7% of their monthly per capita consumption expenditure on different illnesses [15]. Thus, it is revealed that households with low income levels are at risk the most in facing a certain type of a chronic disease due to the heavy financial burden on household and high out of pocket medical expenditures.

In Sri Lanka, Pallegedara [16] examined the effects of chronic NCDs on household's out-of-pocket health expenditures and found that medical poverty is high among chronic NCDs. Pallegedara and Grimm [17] further highlight that older persons are more likely to suffer from chronic diseases. In order to examine the association of NCD-prevalence and healthcare utilization with household consumption, Kumara and Samaratunge [18] employed the two-part model using the 2012/2013 household survey and found private healthcare utilization was negatively related with household consumption. In another study, Kumara and Samaratunge [19] investigated the patterns and determinants of the burden of expenses in household, which found that the burden of expenses does not vary substantially according to the variation in income.

Households and families with chronically-ill members have a higher possibility of encountering health expenditure related poverty than those without. This situation is consistent in both the developed and developing countries [20]. Likewise, along with the substantial percentage growth of chronic patients in households, medical expenditure also shows an increase. Thus, such scenarios shows a tendency to create poverty among such households [14]. Hence, it can be noted that the prevalence of chronic conditions and disabilities can influence, make difference on, both the income and expenditure levels of individuals. Table 1 summarizes some of the common variables used to explain chronic illnesses in social context.

Although there are numerous evidences which indicate rapid growth in chronic diseases, literature is limited to the extent—as to how households experience financial burden that arise due to chronic diseases and disabilities, especially in the Sri Lankan context. Thus, this study will focus to contribute to this research gap by examining the difference of chronic illnesses towards income and expenditure levels of chronically-ill patients in Sri Lanka.

## Data and methodology

### Data

The study is aimed to investigate the differences in the level of income and expenditure among chronically-ill people in the Sri Lankan context. Researchers mainly focused the study based on quantitative data gathered from the latest Household Income and Expenditure Survey (HIES) conducted in 2016 by the Department of Census and Statistics (DCS) in Sri Lanka. The HIES is a sample survey conducted to determine seasonal and regional discrepancies of income, expenditure as well as consumption patterns; the HIES 2016 is the ninth in the series. This survey was held during the period from January to December in 2016 by taking 25,640 housing units into account, covering all 25 districts in the country.

**Table 1. Summary of literature: Variables and supporting research articles.**

| Variable | Past research studies |
|---|---|
| Income | Sultana, Mahumud [4], Burki, Khan [6], Lee, Hamid [7], Essue, Kelly [9], Schoen, Osborn [10], Kemp, Preen [13], Lan, Zhou [14], Pallegedara [16], Pallegedara and Grimm [17], Kumara and Samaratunge [18], Kumara and Samaratunge [19], Goryakin and Suhrcke [20], Almalki, Karami [21], Christopher, Himmelstein [22], Chung, Mercer [23], Fong [24], Kahn, Vest [25], Kim and Richardson [26], Malon, Shah [27], Perruccio, Katz [28], Wang, Sun [29], Bhojani, Thriveni [30], Flores, Krishnakumar [31], Habibov [32] |
| Expenditure | Burki, Khan [6], Lee, Hamid [7], Essue, Kelly [9], Schoen, Osborn [10], Lan, Zhou [14], Pallegedara [16], Pallegedara and Grimm [17], Kumara and Samaratunge [18], Kumara and Samaratunge [19], Goryakin and Suhrcke [20], Bhojani, Thriveni [30], Flores, Krishnakumar [31], Habibov [32] |
| Gender | World Health Organization [1], Sultana, Mahumud [4], Abegunde, Mathers [11], Lan, Zhou [14], Malon, Shah [27], Costa-Font and Gil [33], Jayasinghe [34], Peek, Drum [35] |
| Age | World Health Organization [1], Sultana, Mahumud [4], Almalki, Karami [21], Chung, Mercer [23], Malon, Shah [27], Wang, Sun [29], Habibov [32], Bleich, Koehlmoos [36], Jayasinghe, Selvanathan [37], Lino, Portela [38], Liu, Rao [39], Pati, Agrawal [40] |
| Educational level | Lan, Zhou [14], Almalki, Karami [21], Chung, Mercer [23], Malon, Shah [27], Bailey, Doyle [41], Parodi, Parodi [42] |
| Marital status | Sultana, Mahumud [4], Lan, Zhou [14], Almalki, Karami [21], Wang, Sun [29], August and Sorkin [43], Jayathilaka, Selvanathan [44] |
| Employment status | Sultana, Mahumud [4], Chung, Mercer [23], Malon, Shah [27], Bambra, Whitehead [45], Nazarov, Manuwald [46], Zhang, Zhao [47] |
| Ethnicity and Religion | Murphy, Mahal [5], Abegunde, Mathers [11], Bloom, Chen [12], Bailey, Doyle [41], Arrey, Bilsen [48], Coats, Downey [49], Druedahl, Yaqub [50], Nguyen, Paul [51], Shavers, Bakos [52] |

Source: Authors' compilation.

The survey questionnaire mainly concentrates on three major criteria; demographic characteristics, household expenditure spent on food and non-food, and household income earned in monetary and non-monetary terms. The design of the study is based on two stage stratified sampling. In two stage stratified sampling, the sample population is segregated into different stratas depending on various characteristics such as age, income, geography etc. Here, the main domain used for stratification is the district whereas, Urban, Rural and Estate sectors in each district are the selection domains [53].

## Analytical tool

This study used Analysis of Variance (ANOVA) which is introduced by Fisher [54] and later developed by many statisticians. ANOVA can be used as an exploratory tool to explain observations that assesses potential differences in a scale-level dependent variable by a nominal-level variable having two or more categories. ANOVA is a highly useful method, as it allows the assessment of the influence of some controlled factors on experimental results. The analysis of variance can be carried out according to different schemes [55]. However, the results of the ANOVA are invalid if the independence assumption is violated. In general, with violations of homogeneity the analysis is considered robust if the study have equal sized groups. With violations of normality, continuing with the ANOVA is acceptable if studies are determine a large sample size [56, 57]. One-way ANOVA evaluates the impact of a sole factor on a sole response variable; it is used to determine whether there are any differences that are statistically significant between the means of three or more independent (unrelated) groups. A two-way ANOVA is an extension of the one-way ANOVA. With a two-way ANOVA, two independent groups observe the interaction between the two factors and test the effect of two factors simultaneously [58–60].

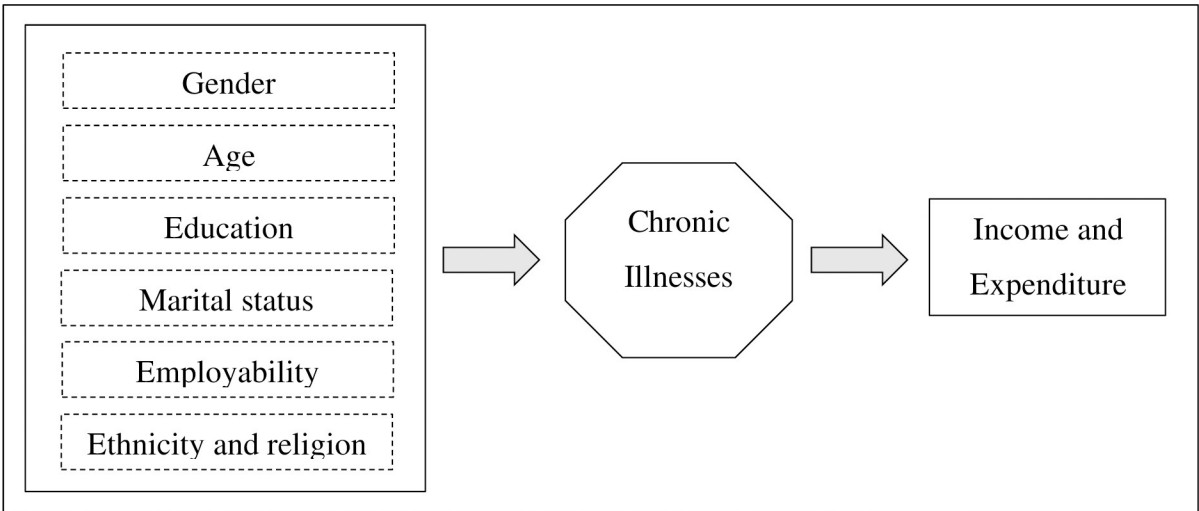

**Fig 1. Conceptual framework.** Source: Authors' compilation.

In conducting the ANOVA test, per capita income along with sources of such income and per capita expenditure along with sources of such expenditure, have been considered as the "Response Variable". All chronic illnesses which are specified previously in the study, have been taken into account separately as the "Factor Variable".

The objective of the study requires, examining the differences in the level of income and expenditure among chronically-ill people. Hence, an ANOVA test has been carried out by taking into account the mean values of both the per capita income and expenditure with regards to chronic illnesses. Initially one-way ANOVA was used to ascertain whether the income and expenditure levels have been varied among illnesses. The results lead the study in using two-way ANOVA. Therefore, in order to investigate the factors that make a significant impact, a two-way ANOVA test was conducted by the researchers [61, 62]. In this model, according to what the name suggests, two factor variables are evaluated against one response variable.

In this study, chronic illnesses have been considered as a whole, pairing up with each of the other variables separately such as gender, age, education level, employability, marital status, religion and ethnicity as factor variables, whereas the mean per capita income and expenditure as the response variable (Fig 1). Hence, the combined effect of the two factor variables on the response variable has been taken into account. This can also be identified as the "Interaction Effect" [62–65]. The results of the two-way ANOVA test are critically analyzed and explained further in the results and discussion section of this study. All computations were tested using STATA 12.0.

This study carries hypothesis to identify whether there are any differences in the level of income and expenditure among chronically people in Sri Lanka. To examine the effects of the status of chronic patients on income and expenditure, two-way ANOVA was performed.

The level of income earned by one household may differ to that of another. At the same time, their expenditure patterns are likely to vary from one to another based on their earnings. A chronically-ill household with a high income will find it less penalizing to allocate a proportion of income towards health care expenses. On the contrary, a household with a basic income may be compelled to cut down expenses related to satisfying their basic needs and wants (such as for food, shelter and etc.). Instead, spend these to acquire health care services. Hence, it is hypothesized that the level of income and expenditure does difference on the poverty of chronically-ill people.

## Results and discussion

The prime objective of the study (i.e. to investigate the differences in the level of income and expenditure of people diagnosed with chronic illnesses) is attained with evidence from the household survey data conducted by the DCS and the Ministry of National Policies and Economic Affairs. The study was based on 25,640 housing units encompassing the entire 25 districts in the country. According to the survey results, out of 82,961 individuals living in the housing units under review, a total of 11,798 individuals suffer from a certain type of chronic disease. It can be assumed that 14.22% of persons are deemed to be victims of chronic illnesses whilst 85.78% can be free from chronic illnesses in Sri Lanka. The health goal 'SDG 3' pertaining to the SDG profile of Sri Lanka issued by the WHO South East Asia Region in 2017 indicates that the likelihood to die from NCDs before an individual reaches the age of 70 is 17.7%. This number is expected to accelerate in the coming years [2]. Thus, the difference of chronic conditions on households is not negligible and therefore, cannot be undervalued.

Moreover, a higher percentage of patients has been recorded in the urban sector which accounted for 16.38%. Western, Central and Southern provinces accounted for more than half of the chronic patients in the population. In the discussion regarding socio-economic characteristics of the population; gender, age, level of education, employability, marital status, income levels and expenditure levels are taken into account, out of which, levels of income and expenditure are considered to be the core area of the study. When considering the level of education, a majority of chronically-ill patients have received tertiary education, out of which, nearly 50% were unemployed due to the disability associated with the respective chronic illness (Table 2).

## Burden of chronic illnesses on income and expenditure of households

In evaluating the level of income, per capita income and total income of the household per month were used as measurements. When analyzing the difference of chronic illnesses towards per capita income, it was identifiable that mean values of per capita income vary along with the disease (Fig 2). Descriptive statistics related to mean per capita income across chronic illnesses depict that the highest mean and standard deviation were recorded in relation to diabetes (mean = 20802.86, SD = 28728.57), while mental retardation recorded the lowest mean per capita income of 10539.185.

In investigating the differences of the mean per capita income and total mean income in the sub-dimensions of chronic illnesses, deployed one-way ANOVA test as the tool. The results generated from ANOVA tests, mean values and standard deviation of the income measurements are depicted in Table 3. The results reveal that patients of each chronic illness earn different levels of mean per capita income (F-value = 22.10; p<0.0001) therefore, can affect the total mean income (F-value = 10.54; p<0.0001). Thus, chronic illnesses have a statistically proven significant difference towards levels of income in the population.

Income is generated via seven sources according to data obtained from the DCS. Hence, it is necessary to identify as to which income level has a significant difference from chronic illnesses. One-way ANOVA tests were conducted to further clarify the significance towards the sources of income; employment income, agricultural income, non-agricultural income, other income, ad hoc income and non-monetary income from food and non-food expenditure (See S1 Appendix). The results of this study indicate that chronic patients earn different levels of total mean income; where mean per capita income is influenced by the above-mentioned sources (p<0.0001). Thus, this study reveals that even though most of the chronic patients were earlier found to be high income earners, the chronic condition and its consequences have significantly affected their level of income. Further, it has proved the fact that chronic diseases

**Table 2. Demographic and socio-economic factors of chronically-ill patients.**

| Demographic and socio-economic characteristics | Population (%) | | Head of the households (%) | |
|---|---|---|---|---|
| | Chronically-ill patients | Not chronically-ill patients | Chronically-ill patients | Not chronically-ill patients |
| **Gender** | | | | |
| Male | 41.56 | 47.70 | 67.21 | 76.57 |
| Female | 58.44 | 52.30 | 32.79 | 23.43 |
| **Age** | | | | |
| 0–14 | 4.03 | 29.11 | 0.00 | 0.01 |
| 15–25 | 2.37 | 16.10 | 0.12 | 1.06 |
| 25–39 | 8.00 | 22.19 | 5.39 | 24.55 |
| 40–65 | 54.34 | 27.04 | 58.03 | 60.83 |
| Above 65 | 31.25 | 5.57 | 36.46 | 13.55 |
| **Ethnicity** | | | | |
| Sinhalese | 69.95 | 73.52 | 73.34 | 72.24 |
| Sri Lankan Tamil | 15.92 | 14.08 | 14.20 | 15.37 |
| Indian Tamil | 3.99 | 3.17 | 3.25 | 3.75 |
| Sri Lankan Moors | 9.77 | 8.76 | 8.63 | 8.30 |
| Malay | 0.21 | 0.25 | 0.30 | 0.19 |
| Burgher | 0.10 | 0.18 | 0.21 | 0.11 |
| Other | 0.06 | 0.04 | 0.07 | 0.04 |
| **Religion** | | | | |
| Buddhism | 69.04 | 66.39 | 68.89 | 68.55 |
| Hinduism | 13.77 | 16.20 | 13.84 | 15.81 |
| Islam | 8.93 | 9.96 | 8.85 | 8.49 |
| Catholic/Christian | 8.26 | 7.44 | 8.42 | 7.12 |
| Other | 0.01 | 0.01 | 0.00 | 0.02 |
| **Education level** | | | | |
| No schooling | 6.58 | 11.73 | 4.57 | 3.02 |
| Primary education | 27.28 | 21.36 | 27.67 | 21.10 |
| Secondary education | 22.64 | 19.10 | 23.34 | 22.45 |
| Tertiary education | 40.95 | 45.11 | 41.30 | 50.51 |
| Higher education | 2.47 | 2.65 | 3.07 | 2.88 |
| Special education | 0.09 | 0.06 | 0.04 | 0.03 |
| **Marital status** | | | | |
| Unmarried | 10.71 | 48.64 | 2.30 | 2.19 |
| Married | 66.99 | 45.42 | 66.87 | 81.42 |
| Widowed | 20.11 | 4.63 | 27.51 | 13.02 |
| Divorced | 0.53 | 0.31 | 0.72 | 0.63 |
| Separated | 1.65 | 1.00 | 2.60 | 2.73 |
| **Employability** | | | | |
| Engaged in economic activity | 36.03 | 37.15 | 50.41 | 0.01 |
| Seeking work | 1.30 | 3.12 | 0.65 | 76.32 |
| Student | 0.97 | 8.04 | 0.02 | 0.62 |
| Household activities | 26.68 | 16.90 | 13.72 | 0.06 |
| Retired | 6.45 | 1.16 | 9.68 | 11.39 |
| Unable to work | 22.97 | 3.74 | 23.89 | 3.51 |
| Other | 1.57 | 0.77 | 1.62 | 7.20 |
| None | 4.03 | 29.11 | 0.00 | 0.89 |
| **Employment status** | | | | |

*(Continued)*

**Table 2.** (Continued)

| Demographic and socio-economic characteristics | Population (%) | | Head of the households (%) | |
|---|---|---|---|---|
| | Chronically-ill patients | Not chronically-ill patients | Chronically-ill patients | Not chronically-ill patients |
| Government employee | 3.51 | 62.45 | 4.36 | 7.80 |
| Semi-government employee | 0.98 | 4.61 | 1.55 | 2.44 |
| Private sector employee | 12.53 | 1.19 | 17.75 | 34.66 |
| Employer | 1.20 | 17.74 | 2.21 | 2.08 |
| Own account worker | 15.68 | 0.64 | 24.93 | 29.52 |
| Contributing family worker | 2.86 | 10.81 | 0.57 | 0.43 |
| None | 63.23 | 2.57 | 48.63 | 23.07 |

Source: Authors' calculation based on the HIES (2016).

have a difference towards the income of victims despite the fact of being low income people [6].

Similar to the income aspect, per capita expenditure and total expenditure were used as measurements in the evaluation of levels of expenditure. When considering the mean per capita expenditure, it has varied across each illness (Fig 3). Descriptive statistics associated with the variables result are similar as income components, where the highest mean value was recorded with diabetes (Mean = LKRs.18, 202.92) and lowest in mental retardation.

In investigating the differences between the mean per capita expenditure and total mean expenditure in the sub-dimensions of chronic illnesses, the following results were taken into consideration (Table 4).

The results from the one-way ANOVA tests conducted for the measurement of expenditure reveals that, the patients who suffer from each chronic illness consume different levels of mean per capita expenditure (F-value = 31.56; $p < 0.0001$) and total mean income (F-value = 14.68; $p < 0.0001$). Further clarification of the study envisages that the categories of the expenditure;

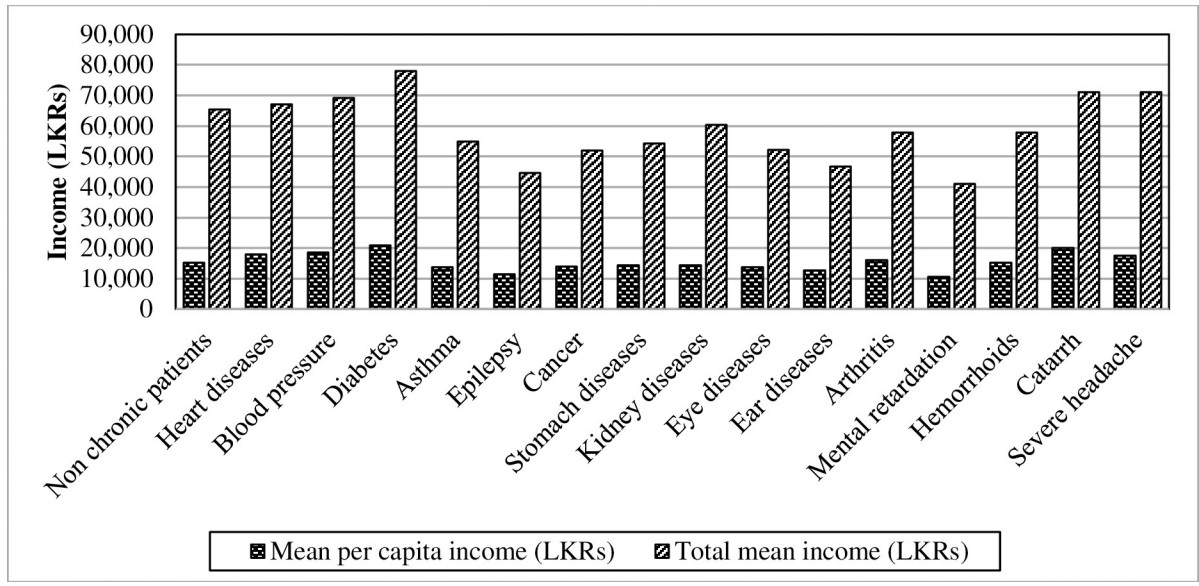

**Fig 2. Mean per capita income and total income of chronically-ill patients.** Source: Authors' illustration based on the HIES (2016).

**Table 3. One-way ANOVA results of the difference of the chronic illnesses towards income (LKR).**

|  | Mean | SD | F | Prob>F |
|---|---|---|---|---|
| Per capita income | 17,489.95 | 24,059.50 | 22.10 | <0.0001 |
| Total income | 65,851.30 | 95,526.69 | 10.54 | <0.0001 |

Source: Authors' calculation based on the HIES (2016).

food expenditure and non-food expenditure have been differenced from chronic illnesses due to the consumption of different levels of expenditure in households.

The food patterns between a chronic patient and a healthy person is considerably different where former seeks healthier food. The burden of having a healthy meal is that the prices of healthy food are high compared to less healthy or unhealthy food [66]. Thus, despite the affordability for healthy food, patients tend to consume healthy meals to maintain the level of the chronic condition, to prevent from further worsening or recover from same. Furthermore, it was identified that 5,283 chronic patients visited government hospitals, out of which, 3,709 patients visited to receive treatments for their illnesses. In addition, 3,175 chronic patients visited private hospitals, out of which, 2,514 patients have visited to receive relevant treatments for their illnesses. This proves the fact that health expenditure on chronic illnesses has definitely differenced expenditure level of households of victims which is inclusive of non-food expenditure [9].

As mentioned above, it is evident that there is a statistically proven significant difference in the mean length of income and expenditure. Hence, there is a significant difference of chronic illnesses towards income and expenditure measurements. If there is any interaction effect with other demographic and socio-demographic factors as discussed above and although the difference could be identified, the main effect of chronic illnesses towards income and expenditure can be misinterpreted. Thus, two-way ANOVA tests were conducted to examine the interaction of chronic illnesses and demographic and socio-economic characteristics of chronically-ill patients on income and expenditure of the victim households in Sri Lanka.

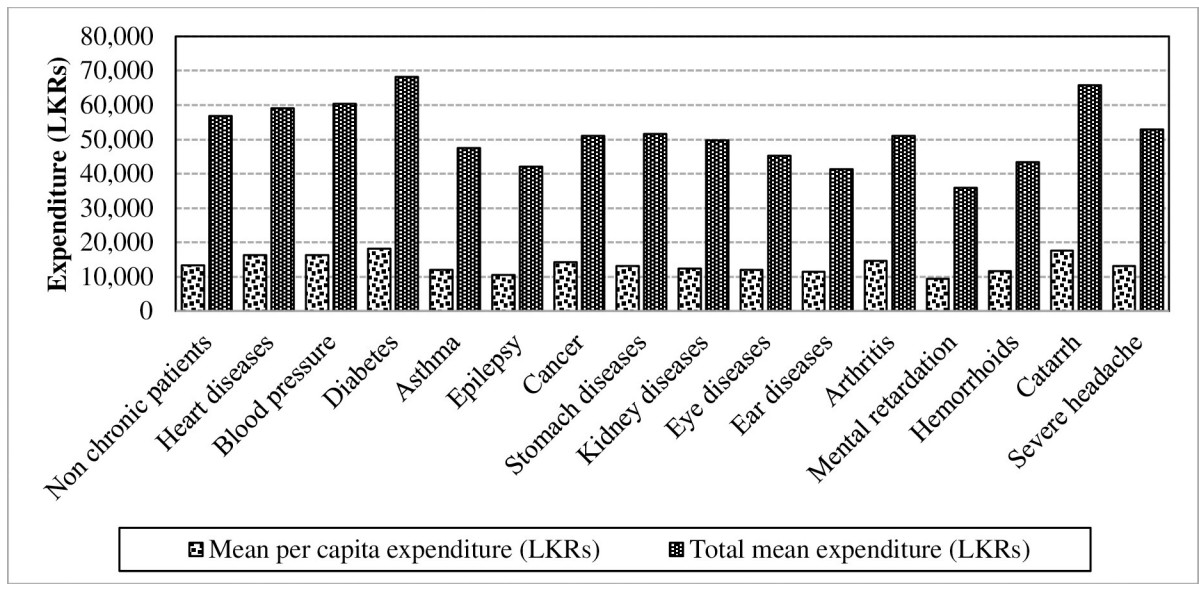

**Fig 3. Mean per capita and total mean expenditure of chronic patients.** Source: Authors' illustration based on the HIES (2016).

**Table 4. One-way ANOVA results of the difference of the chronic illnesses towards expenditure (LKRs).**

|  | Mean | SD | F | Prob>F |
|---|---|---|---|---|
| Per capita expenditure | 15,457.61 | 18,036.25 | 31.56 | <0.0001 |
| Total expenditure | 57,648.50 | 68,751.37 | 14.68 | <0.0001 |

Source: Authors' calculation based on the HIES (2016).

Table 5 shows the main results of the ANOVA tests on mean per capita income. In two-way ANOVA tests, it was revealed that chronic illnesses (F-value = 117.67; p<0.0001) and gender (F-value = 11.14; p = 0.0008) have a statistically proven significant difference towards the mean per capita income. This is in the absence of the interaction effect, while there is a significant interaction between chronic illnesses and gender towards the mean per capita income (F-value = 4.93; p = 0.0205). Thus, the gender characteristic strengthens the difference of per capita income between chronic patients and non-chronic people by 0.0205.

Moreover, Fig 4 depicts as to how the gender characteristic has an effect the mean per capita income of chronically-ill population. Accordingly, it is identifiable that the interaction effect is significant due to the higher impact from male patients.

Fig 5 shows that chronic illnesses do not have a significant effect towards the mean per capita income (F-value = 0.38; p = 0.5382), while the effect of age levels to mean per capita income is statistically significant (F-value = 37.06; p<0.0001) in the absence of the interaction effect. Besides, the mean values of per capita income plotted in Fig 5 illustrates that age levels amplify the difference of chronic illnesses towards per capita income at the significant level 0.01 (F-value = 6.97; p<0.0001).

Moreover, according to the mean of per capita income for the four age groups of chronically-ill population and non-chronic population in Fig 5, the age groups of 25 to 39 years and 40 to 65 years have mainly caused to have an interaction effect followed by the rest of the age groups. In comparison to other demographic and socio-economic characteristics of chronically-ill population, educational level has a stronger impact towards the mean per capita income with a large F-value of 604.97 in the absence of the interaction. Furthermore, there exists a strong tendency for interaction between chronic illnesses and educational level on the mean per capita income (F-value = 30.44; p<0.0001). The interaction effect is highly visualized in the "Secondary", "Tertiary" and "Higher" categories as depicted below while causing to build an interaction (Fig 6).

In the chronically-ill population, mean per capita income differed significantly among different marital statuses. Thus, the impact of interaction between chronic illnesses and marital status is statistically significant (F-value = 2.58; p = 0.0356) as depicted in Fig 7. It depicts that marital status strengthens the differences in per capita income levels between the two tested populations. Moreover, the "Divorced" category was the main cause to amplify the effect. In contrast, chronic illnesses do not imply a significant effect towards the mean per capita income, although marital status has a significant impact in the absence of the interaction effect.

Employability has exhibited a statistically proven significance towards the mean per capita income in the absence of the interaction effect in the chronically-ill population. Thus, it inferred that there exists a significant interaction between chronic illness and employability towards the mean per capita income. The impact of interaction is caused mainly due to the "Unemployed" category as depicted in Fig 8. Most chronic patients who suffer severe conditions from the respective disease have been unemployed [27] and this caused to have an interaction effect.

**Table 5. Two-way ANOVA results of the effect of chronic illnesses with the interaction of demographic and socio-economic factors towards the mean per capita income.**

| Demographic and socio-economic characteristic | Significance of chronic illness | Significance of demographic and socio-economic characteristic | Interaction effect |
|---|---|---|---|
| Gender | <0.0001*** | 0.0008*** | 0.0265** |
| Age level | 0.5382 | <0.0001*** | <0.0001*** |
| Educational level | 0.0051*** | <0.0001*** | <0.0001*** |
| Marital status | 0.8127 | <0.0001*** | 0.0356** |
| Employability | <0.0001*** | <0.0001*** | <0.0001*** |
| Ethnicity | 0.0160** | <0.0001*** | 0.0758* |
| Religion | 0.0435** | <0.0001*** | 0.0255** |

Note: *** Significant at level 1%

** significant at level 5%

* significant at level 10%.

Source: Authors' calculation based on the HIES (2016).

Concerning the ethnicity, there is a lessor significant interaction effect of chronic illnesses and ethnicity towards the mean per capita income. This is despite ethnicity making a significant difference on the mean per capita income in the absence of the interaction effect. In contrast, religion has a higher significant interaction effect with chronic illnesses in the mean per capita income while having a significant impact towards the mean per capita income in the absence of the interaction effect (Table 6). Thus, religion weakens the difference of chronic illnesses towards per capita income.

The results conclude that moderate variables of gender, age, educational level, marital status, employability and religion have a significant difference towards the mean per capita income of chronically-ill households of Sri Lanka (Table 7).

ANOVA tests reveal that, gender is less significant towards the mean per capita expenditure in the absence of the interaction between chronic illnesses and gender at an alpha level of 0.1

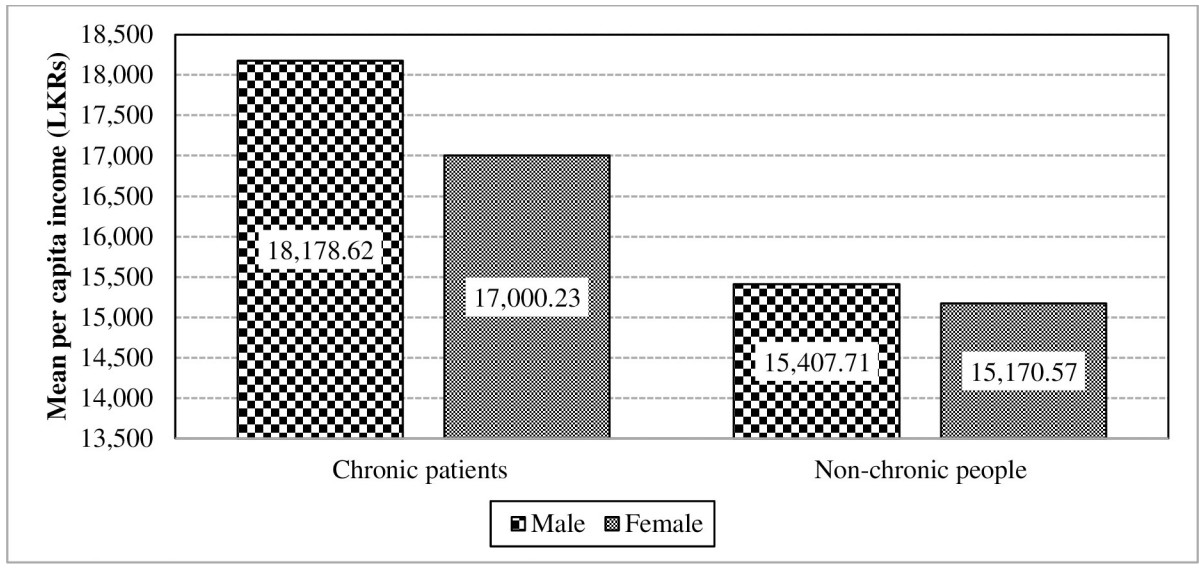

**Fig 4. Interaction of chronic illnesses and gender towards the mean per capita income.** Source: Authors' illustration based on the HIES (2016).

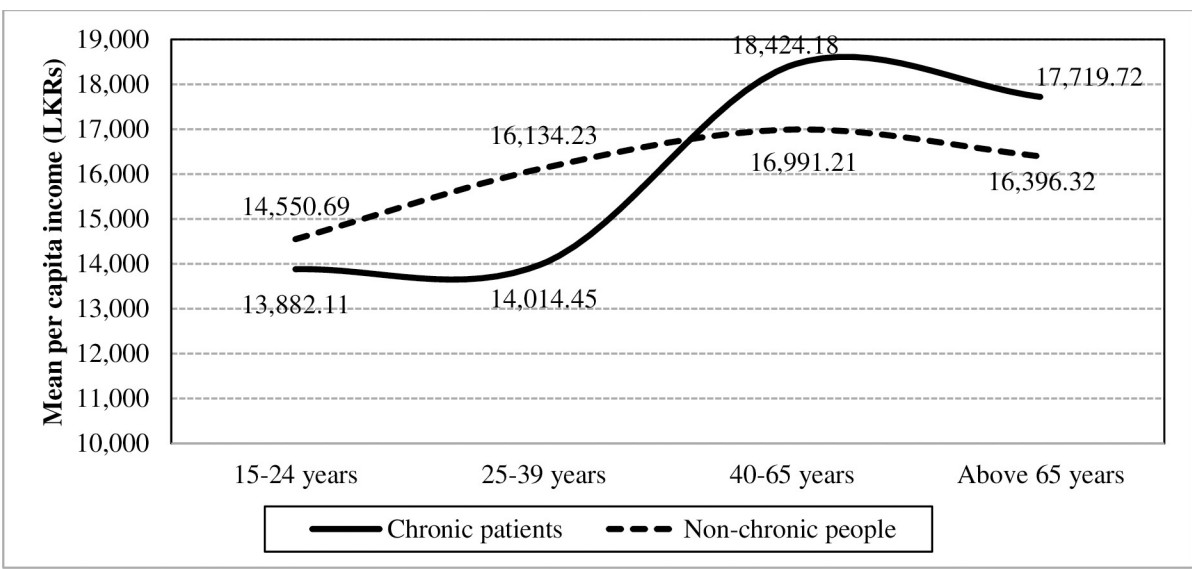

**Fig 5. Interaction of chronic illnesses and age towards the mean per capita income.** Source: Authors' illustration based on the HIES (2016).

(F-value = 3.73; p = 0.0534). Similar to the situation of mean per capita income, gender does have a significant interaction with chronic illnesses towards the mean per capita expenditure. Moreover, the means of per capita expenditure for male and female categories of chronically-ill patients are plotted in the left side of Fig 9. It reveals that male patients magnifies the interaction impact between chronic illnesses and gender towards per capita expenditure. Thus, it infers that gender diversification has moderately strengthen the effect of chronic illnesses towards the mean per capita expenditure.

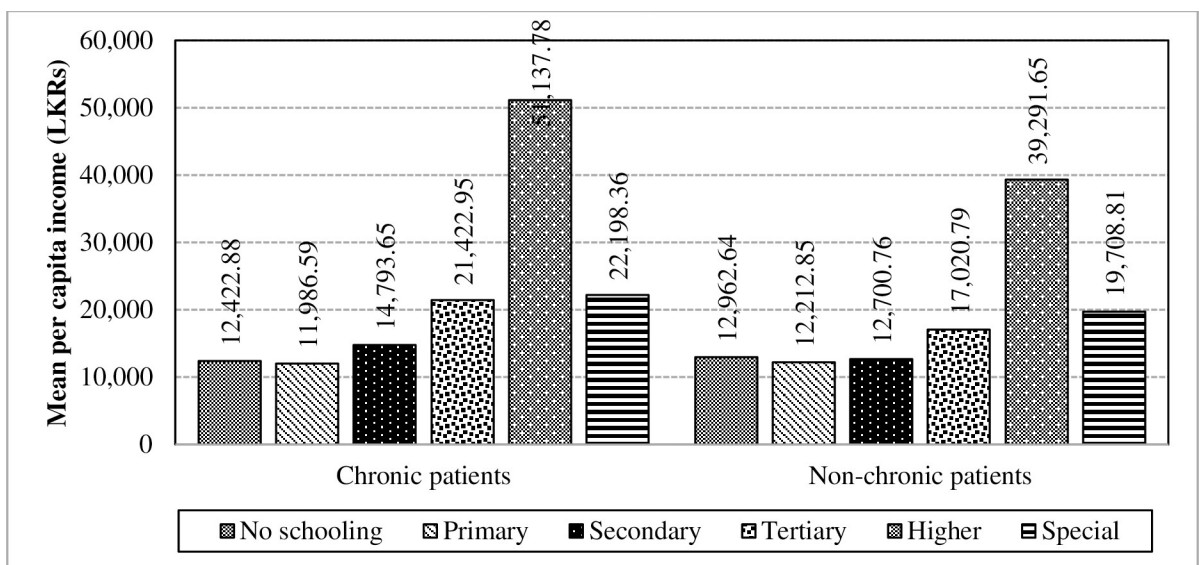

**Fig 6. Interaction of chronic illnesses and educational level towards the mean per capita income.** Source: Authors' illustration based on the HIES (2016).

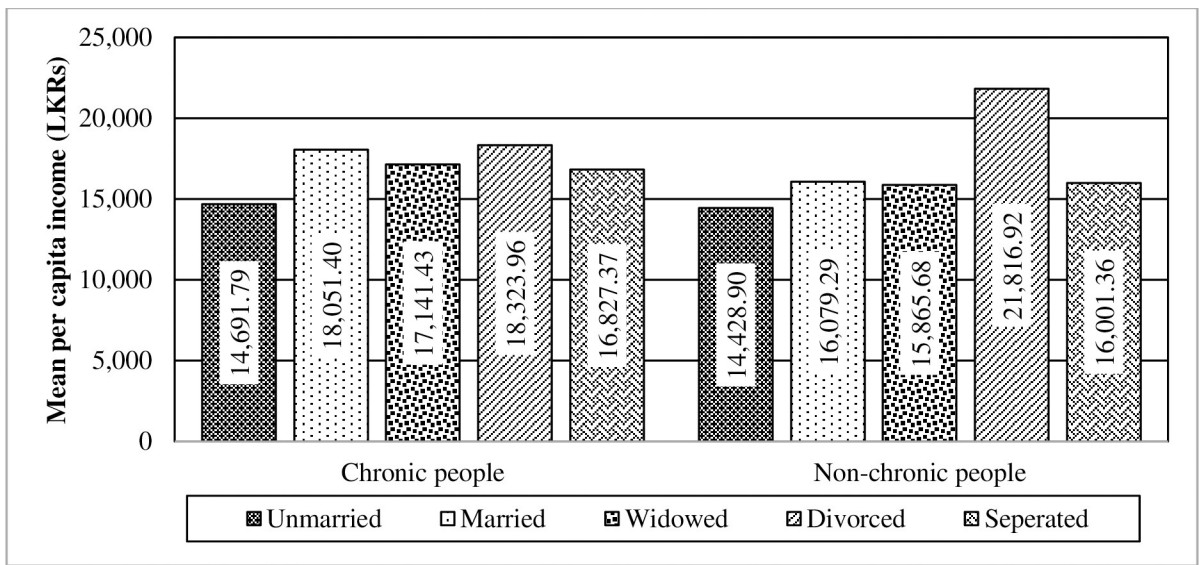

**Fig 7. Interaction of chronic illnesses and marital status towards the mean per capita income.** Source: Authors' illustration based on the HIES (2016).

Results of demographic factors of chronically-ill people disclose that adults who are aged between 40 to 65 years suffer more from chronic illnesses in Sri Lanka (Fig 10). Such a setting has caused to have an interaction which strengthen the interaction impact of chronic illnesses and gender towards the mean per capita expenditure (F-value = 5.76; p = 0.0001. Furthermore, in the absence of the interaction effect, age levels have a statistically proven significant difference towards the mean per capita expenditure (F-value = 33.96; p<0.0001).

A strong tendency is noticeable for the interaction between chronic illnesses and educational level which magnified the difference on the mean per capita expenditure (F-

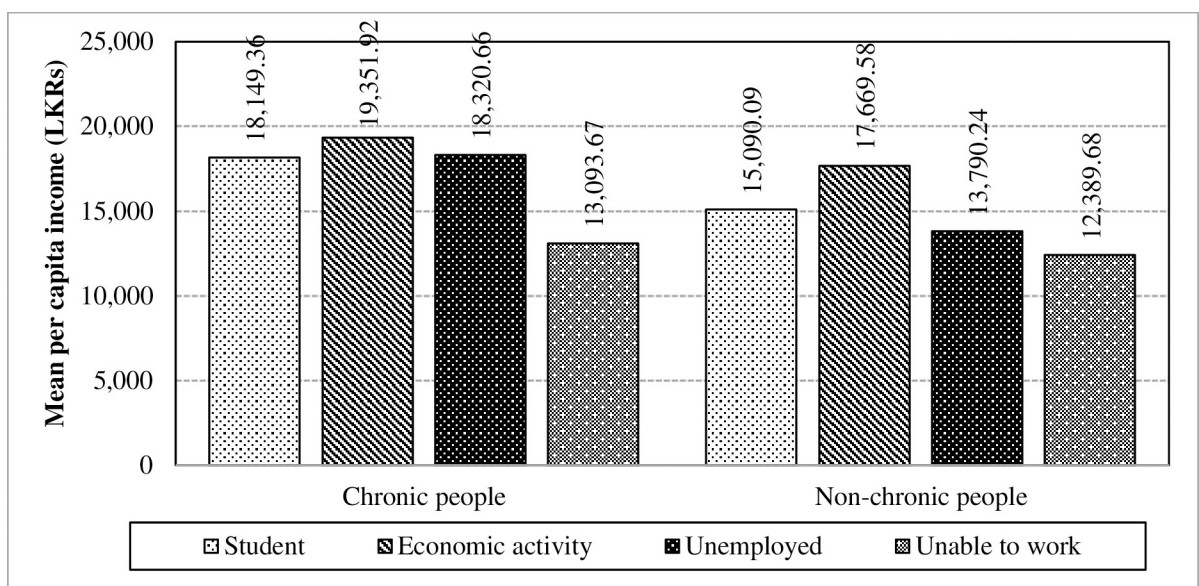

**Fig 8. Interaction of chronic illnesses and employability towards the mean per capita income.** Source: Authors' illustration based on the HIES (2016).

**Table 6. Interaction of chronic illnesses and religion towards the mean per capita income.**

| Source | Analysis of variance | | | | |
|---|---|---|---|---|---|
| | SS | Df | MS | F | Prob>F |
| Model | 5.5058e+11 | 9 | 6.1176e+10 | 139.73 | 0.0000 |
| Chronic patients | 1.7853e+09 | 1 | 1.7853e+09 | 4.08 | 0.0435 |
| Religion | 2.0718e+11 | 4 | 5.1796e+10 | 118.31 | 0.0000 |
| Chronic patients#religion | 4.8602e+09 | 4 | 1.2150e+09 | 2.78 | 0.0255 |
| Residual | 3.6317e+13 | 82951 | 437814255 | | |
| Total | 3.6868e+13 | 82960 | 444403482 | | |

Source: Authors' calculation based on the HIES (2016).

value = 61.74; p<0.0001). Additionally, educational level has a strong difference towards the mean per capita expenditure in the absence of the interaction which has caused to have an interaction effect. The interaction effect is visible in Fig 11 where "No schooling", "Secondary" and "Higher" educational levels have caused to have a higher significant interaction. As such, it reveals that the higher education category has higher mean values in both the chronically-ill and non-chronically-ill population, while the special education category has a lower mean value.

Statistics prove that chronic illnesses (F-value = 7.39; p = 0.0066) and marital status (F-value = 21.22; p<0.0001) have a significant effect towards the mean per capita expenditure in the absence of the interaction effect. Further, the pattern between chronically-ill and non-chronically-ill population has changed due to "Married", "Widowed" and "Divorced" categories (Fig 12). As such, there is a significant difference in the mean values of per capita expenditure between chronic illnesses and marital status, strengthening the interaction impact on expenditure (F-value = 6.91; p<0.0001).

Table proves that employability is one of the factors which amplifies the difference of per capita expenditure between chronic patients and non-chronic people. Thus, it can be concluded that a significant interaction exists between chronic illness and employability towards the mean per capita expenditure. The impact of interaction is caused mainly due to the unemployed category as illustrated in Fig 13. Furthermore, employability has showed a statistically

**Table 7. Two-way ANOVA results of the effect of chronic illnesses with the interaction of demographic and socio-economic factors towards the mean per capita expenditure.**

| Demographic and socio-economic characteristic | Significance of chronic illness | Significance of demographic and socio-economic characteristic | Interaction effect |
|---|---|---|---|
| Gender | <0.0001*** | 0.0534* | 0.0120** |
| Age level | 0.0680* | <0.0001*** | 0.0001*** |
| Educational level | 0.0550* | <0.0001*** | <0.0001*** |
| Marital status | 0.0066*** | <0.0001*** | <0.0001*** |
| Employability | <0.0001*** | <0.0001*** | <0.0001*** |
| Ethnicity | 0.0114** | <0.0001*** | 0.1557 |
| Religion | 0.0027*** | <0.0001*** | <0.0001*** |

Note: *** Significant at 1% level

** significant at 5% level and

* significant at 10% level.

Source: Authors' calculation based on the HIES (2016).

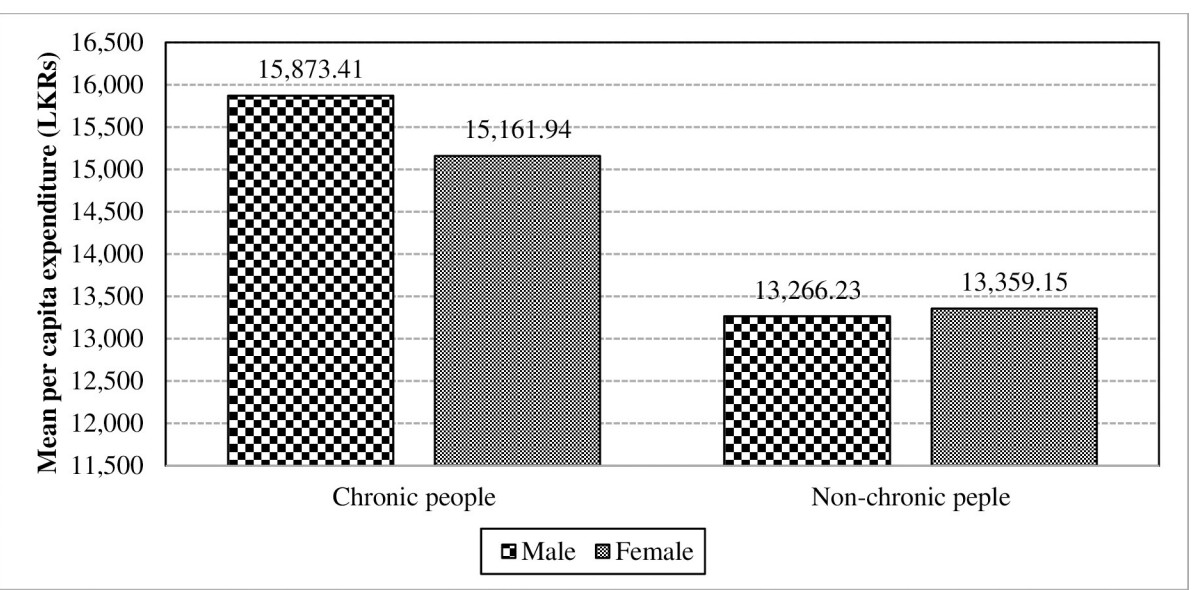

**Fig 9. Interaction of chronic illnesses and gender towards the mean per capita expenditure.** Source: Authors' illustration based on the HIES (2016).

proven significance towards the mean per capita expenditure in the absence of the interaction effect in the chronically-ill population. Chronic illnesses can lead to people being unemployed and subsequently, has caused to have an interaction effect due to the increment in health expenditure along with other related expenses.

Similar to the impact towards the mean per capita income, ethnicity has a significant difference towards the mean per capita expenditure in the absence of the interaction effect (F-value = 88.48; p<0.0001). However, the interaction effect was found non-significant in both

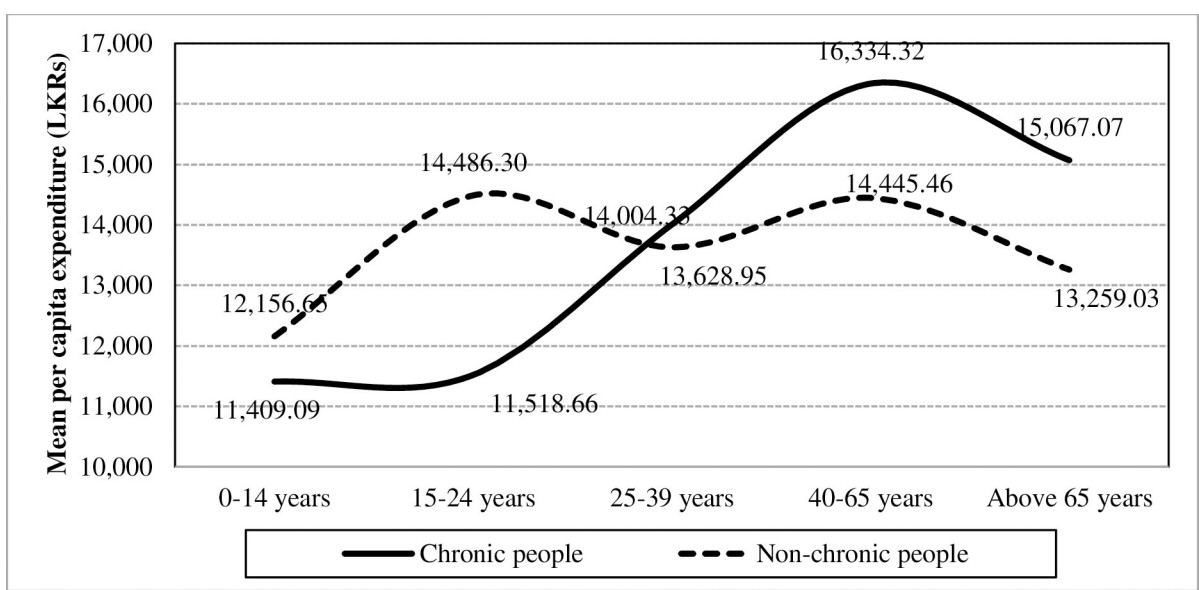

**Fig 10. Interaction of chronic illnesses and age levels towards the mean per capita expenditure.** Source: Authors' illustration based on the HIES (2016).

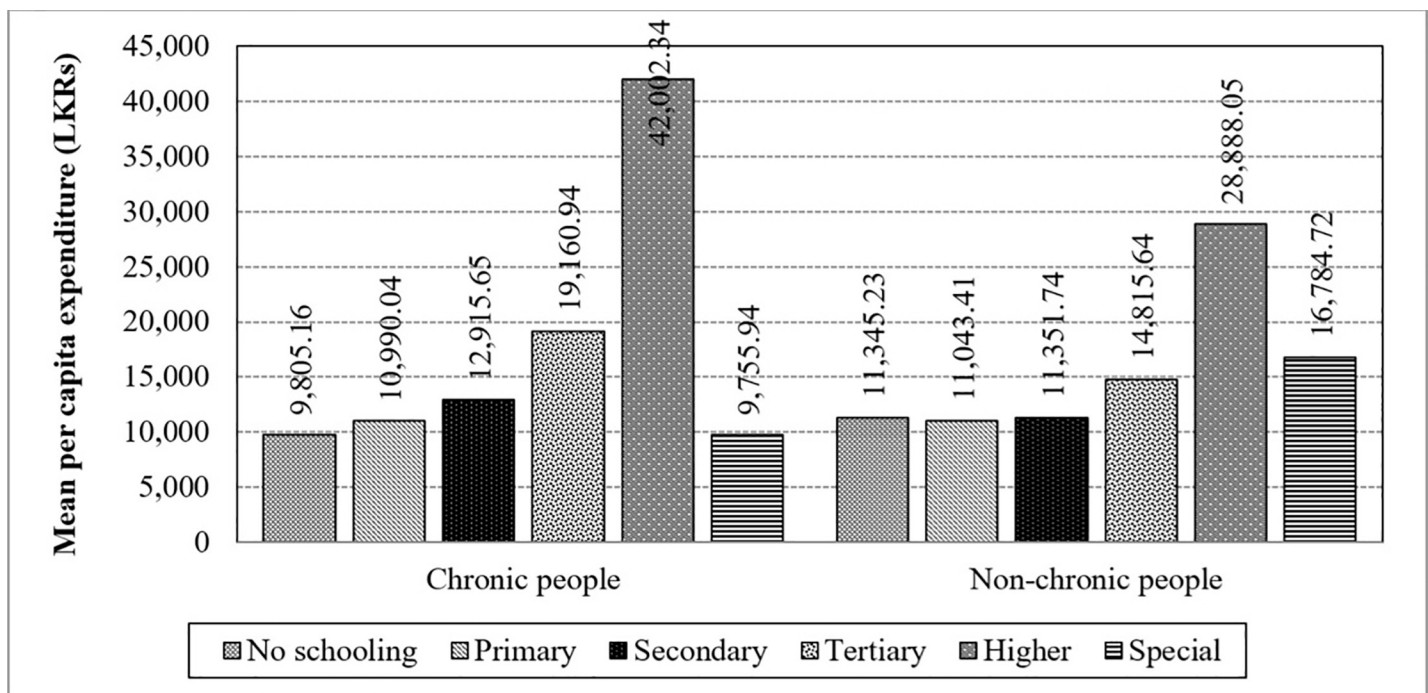

**Fig 11. Interaction of chronic illnesses and educational levels towards the mean per capita expenditure.** Source: Authors' illustration based on the HIES (2016).

the alpha levels of 0.05 and 0.1 (F-value = 1.56; p = 0.1557). Moreover, religion has an interaction effect with chronic illnesses towards the mean per capita expenditure (F-value = 129.72; p<0.0001) while having a significant difference towards the mean per capita expenditure (F-value = 6.35; p<0.0001) in the absence of the interaction effect (Table 8).

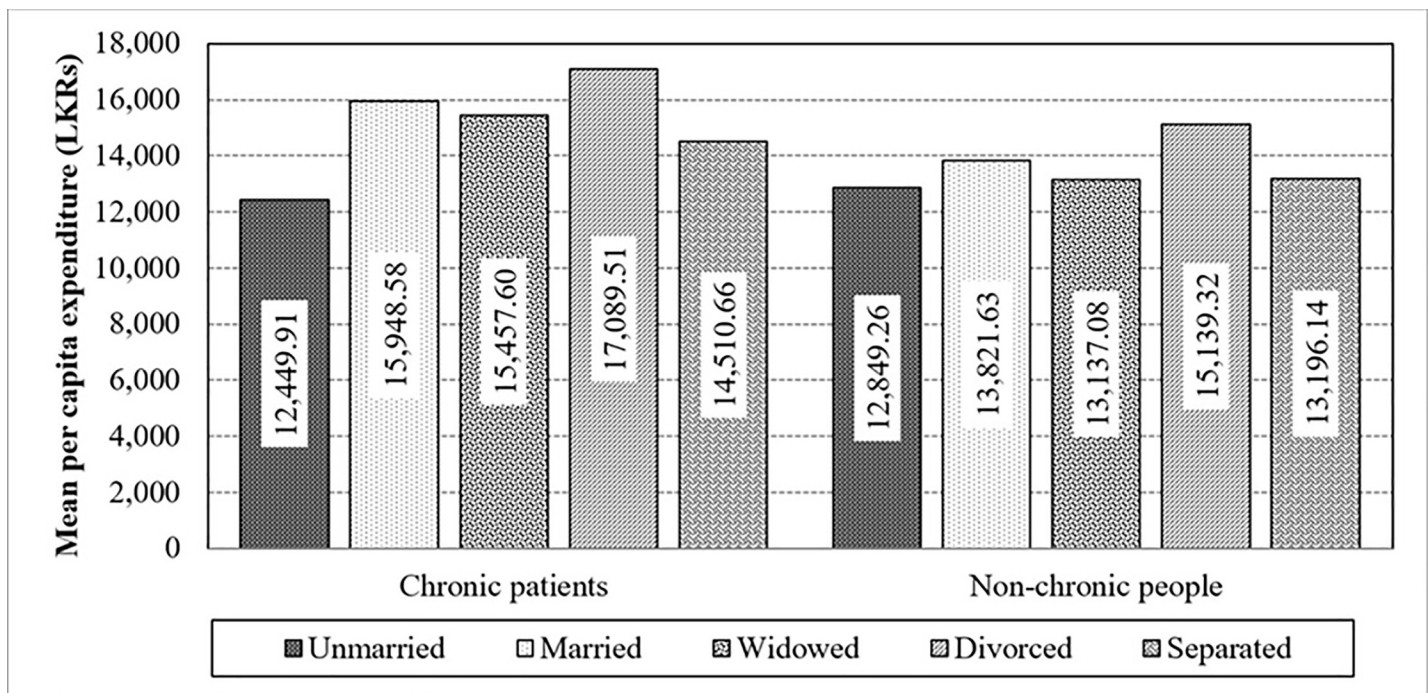

**Fig 12. Interaction of chronic illnesses and marital status towards the mean per capita expenditure.** Source: Authors' illustration based on the HIES (2016).

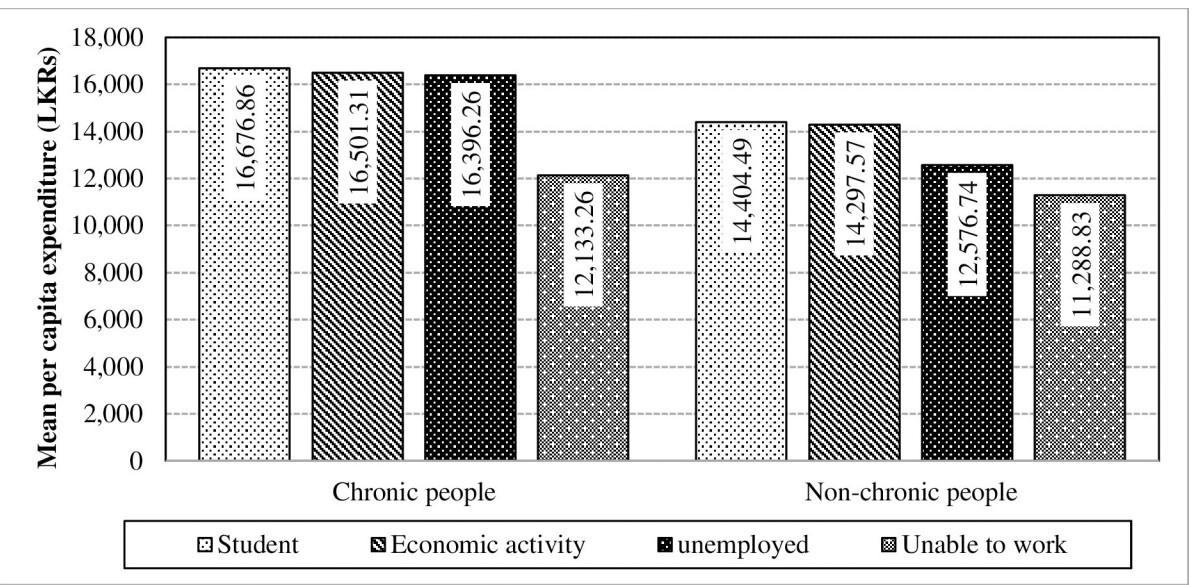

**Fig 13. Interaction of chronic illnesses and employability towards the mean per capita expenditure.** Source: Authors' illustration based on the HIES (2016).

Thus, two-way ANOVA tests infer that the demographic and socio-economic characteristics are considered to have a significant effect on chronic illnesses with regard to variations in the level of income and expenditure among chronically-ill people. On the exception, ethnicity have a significant effect towards mean per capita expenditure of chronically-ill population in Sri Lanka.

## Conclusions and policy implications

The empirical findings of the study reveals evidence gathered by analysis of data deploying ANOVA, from the HIES 2016. This was in order to accomplish the stated prime objective of the research, which was to investigate the differences in the level of income and expenditure among chronically-ill people. It was discovered that married females who do not engage in any type of economic activity, in the age category of 40–65, having an educational level of tertiary education or below and living in the urban sector have a higher likelihood to suffer from chronic diseases. Another fact is that those in 40–65 age category belong to the workforce population. Hence, if they are compelled to be out of employment due to suffering from chronic

**Table 8. Interaction of chronic illnesses and religion towards the mean per capita expenditure.**

| Source | Analysis of variance | | | | |
|---|---|---|---|---|---|
| | **SS** | **Df** | **MS** | **F** | **Prob>F** |
| Model | 3.4705e+11 | 9 | 3.8561e+10 | 154.59 | 0.0000 |
| Chronic patients | 2.2465e+09 | 1 | 2.2465e+09 | 9.01 | 0.0027 |
| Religion | 1.2943e+11 | 4 | 3.2358e+10 | 129.72 | 0.0000 |
| Chronic patients#religion | 6.3397e+09 | 4 | 1.5849e+09 | 6.35 | 0.0000 |
| Residual | 2.0692e+13 | 82951 | 249443833 | | |
| Total | 2.1039e+13 | 82960 | 253600138 | | |

Source: Authors' calculation based on the HIES (2016).

diseases, long term implications on their quality of lives can be severe. These includes forego-ing retirement benefits, ability to have savings, lack of recognition, feeling of insecurity etc. Also, those who are divorced in terms of marital status are more likely to be affected by chronic illness towards per capita income levels. As such, it is reasonable to assume that chronically ill patients who are divorced have no option other than to manage their health expenses by them-selves. In terms of educational level, those in the "No schooling" category are comparatively at a lower position of spending for health care expenses. Moreover, those in the "Higher" educa-tion category spends double the amount (in terms of mean expenditure) of those in the "Ter-tiary" category. Typically, it is assumed that there is hardly any difference between those who have reached "Higher" and "Tertiary" educational levels. Remarkably, these findings challenge some beliefs the society carry with regard to the quality of lives of people in these two educa-tional levels. Under above findings, it can be assumed that most medical expenses are out-of-pocket and income levels can vary under these two categories.

Findings of the study further disclosed that there are significant differences in all variables; mean per capita income, expenditure and in total mean income and expenditure. Further, the analysis discovered that, socio-economic and demographic factors such as age, gender, marital status, educational level, and employability status have a significant effect and a direct relation-ship on chronic illnesses. This is valid in terms of differences in the level of income and expen-diture, except for ethnicity. Latter does not make an effect to the variation of expenditure levels among chronically-ill people.

Moreover, the study infers that chronic illnesses have a statistically proven significant effect towards the income and expenditure level. This has been caused due to the interaction of demographic and socio-economic characteristics associated with chronic illnesses in Sri Lanka.

Study offers some valuable recommendations for decision making on the part of govern-ment which can be highlighted as follows. In 2019, the Government of Sri Lanka budget indi-cates a decline of 1.21% with regard to allocation of healthcare expenditure. However, the magnitude of decline in expenditure (despite percentagewise seems marginal) can be signifi-cant in monetary terms. Thus, it is rational to consider increments on government expenditure on stabilisation and development of healthcare facilities, as an essential factor. In doing so, the Government of Sri Lanka is in a better position to prevent or alleviate chronic illnesses [6]. Having these kind of facilities in place, government can help affected people and families ease their burden of health care expenditure, especially prevent them from falling into poverty. As such, when healthcare policies and private healthcare sector are firm and regulated, it can help handle issues associated with affordability much effectively [36].

It should be stressed that creating private-public sector partnerships and collaborations with the private sector create the potential to devise effective policy instruments in this regard. Contracting out, licensing, franchising, partnerships etc., are some frequent and viable public-private interventions. Moreover, public and private sector collaborations can bring in syner-gies, create channels that are mutual, which can strengthen private sector resources and shar-ing of expertise. More importantly, private public partnerships can help negotiate regulation of pricing policies of private healthcare players. The reason being, typically, private sector health care facilities are considered costly and this keeps many patients away from accessing healthcare services. Nevertheless, this can risk lives of chronic patients for whom receiving continuous medical treatment is crucial. Hence, collaborations can lead towards achieving an effective and affordable service offering that can also enable equitable access for healthcare facilities. By extending licensing and accreditation systems to private healthcare operators, quality of private sector healthcare facilities can be further strengthened. Countries like Brazil, South Africa etc., benefit from successful implementation of such interventions. [67].

Diversifying risks by pooling to a fund for mutual benefit can be proposed as feasible solutions, which can be considered under development of strategies and policies in this regard. This is valid in a context to reduce healthcare expenses associated with persistent diseases such as NCDs, which require continuous treatment. Commercial insurance and community-based mutual services are some practical examples. Those suffering from brain diseases such as, epilepsy, mental retardation and chronic headache as well as cancer and cardiac diseases can immensely benefit from such services. Paving the way for affordable healthcare facilities, developing countries like Colombia, Ghana etc., have implemented such insurance schemes. This can reduce the financial burden, and enhance equitable access to healthcare services.

However, there are certain limitations in this research. The main constraint of this study is limitation of data in the HIES 2016. As such, the list of chronic diseases considered in the survey has not taken into account the other major chronic diseases such as Multiple sclerosis, Parkinson disease and Crohn's disease, as defined by the U.S National Library of Medicine. Obstructive pulmonary is also one of the most common chronic diseases in the world as defined by the WHO that has been ignored in preparing the survey. Hence, this study does not capture the effect of such topical and important chronic diseases which can prevail among the population in the sample under consideration. In addition, the diseases omitted in the study can make a difference, with a more weightage on the level of poverty among chronically-ill people to a certain extent.

In line with the main constraint mentioned above, this study is limited within the scope provided by the HIES 2016. Future studies should expand to incorporate a comprehensive collection of chronically-ill people in Sri Lanka. It can assist to gain valuable insights on overall trends in Sri Lanka's healthcare sector as well as to devise effective polices and mechanisms.

## Supporting information

**S1 Appendix. One-way ANOVA results.**
(DOCX)

## Acknowledgments

The authors would like to thank Dr. (Mrs.) I. R. Bandara, Director General of Department of Census and Statistics, Sri Lanka who gave permission to access Household Income and Expenditure Survey. The authors also would like to thank Ms. Gayendri Karunarathne for proofreading and editing this manuscript.

## Author Contributions

**Conceptualization:** Ruwan Jayathilaka, Sheron Joachim, Venuri Mallikarachchi, Dhanushika Ranawaka.

**Formal analysis:** Ruwan Jayathilaka, Nishali Perera.

**Methodology:** Ruwan Jayathilaka.

**Supervision:** Ruwan Jayathilaka.

**Validation:** Ruwan Jayathilaka, Nishali Perera.

**Writing – original draft:** Ruwan Jayathilaka, Sheron Joachim, Venuri Mallikarachchi, Nishali Perera, Dhanushika Ranawaka.

**Writing – review & editing:** Ruwan Jayathilaka.

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
