## [Decision Letter · Decision Letter 0]

6 Aug 2020

PONE-D-20-08967

Chronic Diseases impacting the income and expenditure of chronically-ill people in Sri Lanka

PLOS ONE

Dear Dr. Jayathilaka,

Thank you for submitting your manuscript to PLOS ONE. After careful consideration, we feel that it has merit but does not fully meet PLOS ONE’s publication criteria as it currently stands. Therefore, we invite you to submit a revised version of the manuscript that addresses the points raised during the review process.

We look forward to receiving your revised manuscript.

Kind regards,

Khurshid Alam, Ph. D.

Academic Editor

PLOS ONE

Journal Requirements:

4. We note you have included a table to which you do not refer in the text of your manuscript. Please ensure that you refer to Table 4, 5, 6, & 7 in your text; if accepted, production will need this reference to link the reader to the Table.

Reviewers' comments:

Reviewer's Responses to Questions

**Comments to the Author**

1. Is the manuscript technically sound, and do the data support the conclusions?

Reviewer #1: Yes

Reviewer #2: Yes

2. Has the statistical analysis been performed appropriately and rigorously? 

Reviewer #1: Yes

Reviewer #2: Yes

3. Have the authors made all data underlying the findings in their manuscript fully available?

Reviewer #1: No

Reviewer #2: Yes

4. Is the manuscript presented in an intelligible fashion and written in standard English?

Reviewer #1: Yes

Reviewer #2: Yes

5. Review Comments to the Author

Reviewer #1: This paper, using ANOVA technique analyses the household income and expenditure (total and per-capita) across a number of chronic disease types. The data used for the analysis is suitable for such analysis and techniques used are reasonable. This paper analyses a unique topic with respect to Sri Lanka, and hence this research makes a significant contribution to the literature.

However, I would like to raise the following concerns that authors need to address.

Introduction and objectives

The problem statement and objective sections have significant repetitions. Unless it is a requirement of the journal, the paper will read better if the authors combine these two sections with the introduction section and remove repetitive sentences.

I have some concerns about the objective of the study.

While in few places, authors claim that they investigate the “differences in the level of income and expenditure of people diagnosed with chronic illnesses”, in the hypothesis they are testing, they note that the hypothesis is that “There is an impact of differences in the level of income and expenditure among chronically-ill people in Sri Lanka”. In my view, authors are not investigating the impact of chronic diseases on income and expenditure levels. The paper analyses the differences in income and expenditure across different types of chronic illnesses. This point needs to be clarified and expressed clearly. In another place, they note that “this research will be carried out by examining the growing toll of chronic diseases and its relevance to poverty”. However, authors neither present any estimated results anywhere how their findings are related to poverty nor how they define poor/non-poor. One way to look into this can be whether the per-capita income/per-capita expenditure in households that have chronically ill people is less than the official poverty line?

Authors present a number of arguments without any references, for example, “According to several investigations conducted, it was revealed that the availability of essential laboratory facilities and drugs for various chronic diseases is limited to a high extent.”

Literature review

While this specific research topic may have not been investigated in Sri Lanka, some literature even remotely related to the context of Sri Lanka, South Asia, etc could be discussed and compared and contrasted with the findings of this paper.

Data

The discussion of the summary statistics is more appropriate to be included in the Data section rather than in the results and discussion section.

Results and discussion

In the results and discussion section, authors note that “Thus, this study reveals that even though most of the chronic patients were found to be non-poor, the chronic condition and its consequences have significantly affected their level of income. Further, it has proved the fact that chronic diseases have an impact towards the income of victims despite the fact of being poor [6]”. Firstly, these two sentences are contradictory. Secondly, I am not sure how the authors relate their findings to poor and non-poor as this paper doesn’t analyse the differences between poor and non-poor.

The authors discuss how chronic diseases may have implications for food and non-food expenditures. For example, they note that "This has caused to have a significant impact of chronic illnesses towards food expenditure." However, they neither provide references for these claims nor provide estimated results in the current study. It would be interesting to know whether there are any such differences in the context of Sri Lanka.

Conclusion

In the introduction, authors note that “this study aims to contribute its findings to policymakers and responsible authorities to devise feasible policies and initiatives.” However, they do not discuss the policy implications of their findings explicitly following the results and discussion section.

Minor comments

Automatic links to some tables are broken.

One-way ANOVA tests were conducted to further clarify the significance towards the sources of income; employment income, agricultural income, non-agricultural income, other income, ad hoc income and non-monetary income from food and non-food expenditure, .

…four out of five chronic disease deaths that occur in the world today spur from low and middle income countries like ours.? Sri Lanka?

Reviewer #2: This manuscript addresses an important topic “Chronic Diseases impacting the income and expenditure of chronically-ill people in Sri Lanka” using Sri Lankan household data. It is useful for understanding of impacts of chronic diseases on household income and expenses among Sri Lankan households. It is of value because Sri Lanka government try to improve health sector in general and introduce several recent health policies such as lowering prices of medicine and restriction of charges by the private hospitals. It is well written, and well structured.

However, I would recommend the authors to consider a number of major revisions for further polishing before publishing in PlosOne.

1. Authors have included several literature related to economic impacts of chronic diseases but several recent papers on Sri Lankan context are missing. For example, Kumara and Samaratunge (2016), Kumara and Samaratunge (2017), Pallegedara and Grimm (2018), Pallegedara (2018) also examined the out-of-pocket health care expenses and welfare impacts due to chronic health diseases in Sri Lanka context. Thus, authors need to add these papers when discussing the related Sri Lankan literature.

1. Kumara, A. S., & Samaratunge, R. (2016). Patterns and determinates of out-of-pocket health care expenditure in Sri Lanka: Evidence from household surveys. Health Policy and Planning, 31(8), 970-983.

2. Kumara, A. S., & Samaratunge, R. (2017). Impact of ill-health on household consumption in Sri Lanka: Evidence from household survey data. Social science & medicine (1982), 195, 68.

3. Pallegedara, A., & Grimm, M. (2018). Have out‐of‐pocket health care payments risen under free health care policy? The case of Sri Lanka. The International journal of health planning and management, 33(3), e781-e797.

4. Pallegedara, A. (2018). Impacts of chronic non-communicable diseases on households’ out-of-pocket healthcare expenditures in Sri Lanka. International Journal of Health Economics and Management. Vol. 18, No. 3. pp. 301-319. DOI: https://doi.org/10.1007/s10754-018-9235-2

2. In this study, authors only describe analytical tools they used. However, they should add a conceptual/analytical framework to explain the choice of variables both independent and dependent variables they used in empirical analysis. Authors can link previous literature regarding the variable selection and should provide more justification for the choice of variables based on the conceptual framework.

3. Authors only used ANOVA method to analyze the data. ANOVA is mainly descriptive tool to explore data. Thus, they need to justify why they used ANOVA over other statistical methods such as regression analysis.

4. Authors did not explain policy implications of their results. Thus, they need to add policy implications and/or recommendations based on the results they found.

6. PLOS authors have the option to publish the peer review history of their article (what does this mean?). If published, this will include your full peer review and any attached files.

Reviewer #1: No

Reviewer #2: No

---

## [Author Response · Author response to Decision Letter 0]

25 Aug 2020

Reviewer 1

Comments to the Author: 

This paper, using ANOVA technique analyses the household income and expenditure (total and per-capita) across a number of chronic disease types. The data used for the analysis is suitable for such analysis and techniques used are reasonable. This paper analyses a unique topic with respect to Sri Lanka, and hence this research makes a significant contribution to the literature. However, I would like to raise the following concerns that authors need to address.

Response from Authors: 

Thank you very much. Well noted.

Comments to the Author: 

Introduction and objectives

The problem statement and objective sections have significant repetitions. Unless it is a requirement of the journal, the paper will read better if the authors combine these two sections with the introduction section and remove repetitive sentences.

Response from Authors: 

Well noted. As suggested, the problem statement and objective sections have been revised. Furthermore, introduction section has been adjusted after removing repetitive sentences.

Comments to the Author: 

I have some concerns about the objective of the study.

While in few places, authors claim that they investigate the “differences in the level of income and expenditure of people diagnosed with chronic illnesses”, in the hypothesis they are testing, they note that the hypothesis is that “There is an impact of differences in the level of income and expenditure among chronically-ill people in Sri Lanka”. In my view, authors are not investigating the impact of chronic diseases on income and expenditure levels. The paper analyses the differences in income and expenditure across different types of chronic illnesses. This point needs to be clarified and expressed clearly. 

In another place, they note that “this research will be carried out by examining the growing toll of chronic diseases and its relevance to poverty”. However, authors neither present any estimated results anywhere how their findings are related to poverty nor how they define poor/non-poor. One way to look into this can be whether the per-capita income/per-capita expenditure in households that have chronically ill people is less than the official poverty line?

Response from Authors: 

Correction has been incorporated as follows.

“This study carries hypothesis to identify whether there are any differences in the level of income and expenditure among chronically people in Sri Lanka. To examine the effects of the status of chronic patients on income and expenditure, two-way ANOVA was performed.”

This correcting has been incorporated in other sections including the title. 

Following sentence has been deleted.

“This research will be carried out by examining the growing toll of chronic diseases and its relevance to poverty, with specific attention to Sri Lanka.”

The word “non-poor” has changed to high income earners. Since, this study did not use any official poverty line to differentiate poor or non-poor people.

Comments to the Author: 

Authors present a number of arguments without any references, for example, “According to several investigations conducted, it was revealed that the availability of essential laboratory facilities and drugs for various chronic diseases is limited to a high extent.”

Response from Authors: 

Reference has been included as follows,

“According to the Ministry of Health and Nutrition and Indigenous Medicine (2), it was revealed that the availability of essential laboratory facilities and drugs for various chronic diseases is limited to a high extent.”

Comments to the Author: 

Literature review

While this specific research topic may have not been investigated in Sri Lanka, some literature even remotely related to the context of Sri Lanka, South Asia, etc could be discussed and compared and contrasted with the findings of this paper.

Response from Authors: 

The following paragraph has been added to the manuscript, which provide details of studies on the subject relating to Sri Lanka.

“In Sri Lanka, Pallegedara (16) examined the effects of chronic NCDs on household’s out-of-pocket health expenditures and found that medical poverty is high among chronic NCDs. Pallegedara and Grimm (17) further highlight that older persons are more likely to suffer from chronic diseases. In order to examine the association of NCD-prevalence and healthcare utilization with household consumption, Kumara and Samaratunge (18) employed the two-part model using the 2012/2013 household survey and found private healthcare utilization was negatively related with household consumption. In another study, Kumara and Samaratunge (19) investigated the patterns and determinants of the burden of expenses in household, which found that the burden of expenses does not vary substantially according to the variation in income.”

Comments to the Author: 

Data

The discussion of the summary statistics is more appropriate to be included in the Data section rather than in the results and discussion section.

Response from Authors: 

Summary statistics are given in Table 2 and is followed by an analysis of the results reported in the table. Therefore, we believe that this section is more appropriate to be included under the results section.

Comments to the Author: 

Results and discussion

In the results and discussion section, authors note that “Thus, this study reveals that even though most of the chronic patients were found to be non-poor, the chronic condition and its consequences have significantly affected their level of income. Further, it has proved the fact that chronic diseases have an impact towards the income of victims despite the fact of being poor [6]”. Firstly, these two sentences are contradictory. Secondly, I am not sure how the authors relate their findings to poor and non-poor as this paper doesn’t analyse the differences between poor and non-poor.

The authors discuss how chronic diseases may have implications for food and non-food expenditures. For example, they note that "This has caused to have a significant impact of chronic illnesses towards food expenditure." However, they neither provide references for these claims nor provide estimated results in the current study. It would be interesting to know whether there are any such differences in the context of Sri Lanka.

Response from Authors: 

These two sentences have been reworded as follows.

“Thus, this study reveals that even though most of the chronic patients were earlier found to be high income earners, the chronic condition and its consequences have significantly affected their level of income. Further, it has proved the fact that chronic diseases have a difference towards the income of victims despite the fact of being low income people [6]”

Confused sentence has been deleted.

“This has caused to have a significant impact of chronic illnesses towards food expenditure.”

Comments to the Author: 

Conclusion

In the introduction, authors note that “this study aims to contribute its findings to policymakers and responsible authorities to devise feasible policies and initiatives.” However, they do not discuss the policy implications of their findings explicitly following the results and discussion section.

Response from Authors: 

Conclusion section has been strengthen by adding few sentences to the first paragraph of the conclusion section. Furthermore, following three paragraphs were added to cover the policy implication and the section has been renamed as “Conclusions and Policy implications.”

“Study offers some valuable recommendations for decision making on the part of government which can be highlighted as follows. In 2019, the Government of Sri Lanka budget indicates a decline of 1.21% with regard to allocation of healthcare expenditure. However, the magnitude of decline in expenditure (despite percentagewise seems marginal) can be significant in monetary terms. Thus, it is rational to consider increments on government expenditure on stabilisation and development of healthcare facilities, as an essential factor. In doing so, the Government of Sri Lanka is in a better position to prevent or alleviate chronic illnesses [6]. Having these kind of facilities in place, government can help affected people and families ease their burden of health care expenditure, especially prevent them from falling into poverty. As such, when healthcare policies and private healthcare sector are firm and regulated, it can help handle issues associated with affordability much effectively [36].

 It should be stressed that creating private-public sector partnerships and collaborations with the private sector create the potential to devise effective policy instruments in this regard. Contracting out, licensing, franchising, partnerships etc., are some frequent and viable public-private interventions. Moreover, public and private sector collaborations can bring in synergies, create channels that are mutual, which can strengthen private sector resources and sharing of expertise. More importantly, private public partnerships can help negotiate regulation of pricing policies of private healthcare players. The reason being, typically, private sector health care facilities are considered costly and this keeps many patients away from accessing healthcare services. Nevertheless, this can risk lives of chronic patients for whom receiving continuous medical treatment is crucial. Hence, collaborations can lead towards achieving an effective and affordable service offering that can also enable equitable access for healthcare facilities. By extending licensing and accreditation systems to private healthcare operators, quality of private sector healthcare facilities can be further strengthened. Countries like Brazil, South Africa etc., benefit from successful implementation of such interventions. [67]. 

 Diversifying risks by pooling to a fund for mutual benefit can be proposed as feasible solutions, which can be considered under development of strategies and policies in this regard. This is valid in a context to reduce healthcare expenses associated with persistent diseases such as NCDs, which require continuous treatment. Commercial insurance and community-based mutual services are some practical examples. Those suffering from brain diseases such as, epilepsy, mental retardation and chronic headache as well as cancer and cardiac diseases can immensely benefit from such services. Paving the way for affordable healthcare facilities, developing countries like Colombia, Ghana etc., have implemented such insurance schemes. This can reduce the financial burden, and enhance equitable access to healthcare services."

Comments to the Author: 

Minor comments

Automatic links to some tables are broken.

One-way ANOVA tests were conducted to further clarify the significance towards the sources of income; employment income, agricultural income, non-agricultural income, other income, ad hoc income and non-monetary income from food and non-food expenditure, .

…four out of five chronic disease deaths that occur in the world today spur from low and middle income countries like ours.? Sri Lanka?

Response from Authors: 

Automatic table links have been corrected

Appendix A has been added to the manuscript as supplementary data. The sentence has been changed as follows.

“….One-way ANOVA tests were conducted to further clarify the significance towards the sources of income; employment income, agricultural income, non-agricultural income, other income, ad hoc income and non-monetary income from food and non-food expenditure (See Appendix A)….”

The sentence has been corrected as follows.

“According to the World Health Organization (1), four out of five chronic disease deaths that occur in the world today spur from low and middle income countries like Sri Lanka.”

Reviewer 2

Comments to the Author: 

This manuscript addresses an important topic “Chronic Diseases impacting the income and expenditure of chronically-ill people in Sri Lanka” using Sri Lankan household data. It is useful for understanding of impacts of chronic diseases on household income and expenses among Sri Lankan households. It is of value because Sri Lanka government try to improve health sector in general and introduce several recent health policies such as lowering prices of medicine and restriction of charges by the private hospitals. It is well written, and well structured.

However, I would recommend the authors to consider a number of major revisions for further polishing before publishing in PlosOne.

Response from Authors: 

Well noted.

Comments to the Author: 

1. Authors have included several literature related to economic impacts of chronic diseases but several recent papers on Sri Lankan context are missing. For example, Kumara and Samaratunge (2016), Kumara and Samaratunge (2017), Pallegedara and Grimm (2018), Pallegedara (2018) also examined the out-of-pocket health care expenses and welfare impacts due to chronic health diseases in Sri Lanka context. Thus, authors need to add these papers when discussing the related Sri Lankan literature.

1. Kumara, A. S., & Samaratunge, R. (2016). Patterns and determinates of out-of-pocket health care expenditure in Sri Lanka: Evidence from household surveys. Health Policy and Planning, 31(8), 970-983.

2. Kumara, A. S., & Samaratunge, R. (2017). Impact of ill-health on household consumption in Sri Lanka: Evidence from household survey data. Social science & medicine (1982), 195, 68.

3. Pallegedara, A., & Grimm, M. (2018). Have out‐of‐pocket health care payments risen under free health care policy? The case of Sri Lanka. The International journal of health planning and management, 33(3), e781-e797.

4. Pallegedara, A. (2018). Impacts of chronic non-communicable diseases on households’ out-of-pocket healthcare expenditures in Sri Lanka. International Journal of Health Economics and Management. Vol. 18, No. 3. pp. 301-319. DOI: https://doi.org/10.1007/s10754-018-9235-2

Response from Authors: 

Comments has been well noted. Kumara and Samaratunge (2016), Kumara and Samaratunge (2017), Pallegedara and Grimm (2018), Pallegedara (2018) have been added to the newly created Table 1.

Furthermore, following paragraph have been added to the manuscript to emphasise Sri Lankan studies.

“In Sri Lanka, Pallegedara (16) examined the effects of chronic NCDs on household’s out-of-pocket health expenditures and found that medical poverty is high among chronic NCDs. Pallegedara and Grimm (17) further highlight that older persons are more likely to suffer from chronic diseases. In order to examine the association of NCD-prevalence and healthcare utilization with household consumption, Kumara and Samaratunge (18) employed the two-part model using the 2012/2013 household survey and found private healthcare utilization was negatively related with household consumption. In another study, Kumara and Samaratunge (19) investigated the patterns and determinants of the burden of expenses in household, which found that the burden of expenses does not vary substantially according to the variation in income.”

Figure 1 and Table 1 are introduced to manuscript, as suggested.

Conceptual /analytical framework has been included as Fig 1 to explain the choice of variables. 

Table 1 included to link the variable selection and justification has been given in the text.

Comments to the Author: 

2. In this study, authors only describe analytical tools they used. However, they should add a conceptual/analytical framework to explain the choice of variables both independent and dependent variables they used in empirical analysis. Authors can link previous literature regarding the variable selection and should provide more justification for the choice of variables based on the conceptual framework.

Response from Authors: 

Figure 1 and Table 1 are introduced to manuscript, as suggested.

Conceptual /analytical framework has been included as Fig 1 to explain the choice of variables. 

Table 1 included to link the variable selection and justification has been given in the text.

Comments to the Author: 3. Authors only used ANOVA method to analyze the data. ANOVA is mainly descriptive tool to explore data. Thus, they need to justify why they used ANOVA over other statistical methods such as regression analysis.

Response from Authors: 

Comment is noted. The following sentences were added to justify why this study used ANOVA rather than other methods.

“….ANOVA is a highly useful method, as it allows the assessment of the influence of some controlled factors on experimental results. The analysis of variance can be carried out according to different schemes [55]. However, the results of the ANOVA are invalid if the independence assumption is violated. In general, with violations of homogeneity the analysis is considered robust if the study have equal sized groups. With violations of normality, continuing with the ANOVA is acceptable if studies are determine a large sample size [56, 57]….”

Comments to the Author: 4. Authors did not explain policy implications of their results. Thus, they need to add policy implications and/or recommendations based on the results they found.

Response from Authors: 

Following three new paragraphs were added to the conclusion section and it is now stated as “Conclusions and Policy implications”.

“Study offers some valuable recommendations for decision making on the part of government which can be highlighted as follows. In 2019, the Government of Sri Lanka budget indicates a decline of 1.21% with regard to allocation of healthcare expenditure. However, the magnitude of decline in expenditure (despite percentagewise seems marginal) can be significant in monetary terms. Thus, it is rational to consider increments on government expenditure on stabilisation and development of healthcare facilities, as an essential factor. In doing so, the Government of Sri Lanka is in a better position to prevent or alleviate chronic illnesses [6]. Having these kind of facilities in place, government can help affected people and families ease their burden of health care expenditure, especially prevent them from falling into poverty. As such, when healthcare policies and private healthcare sector are firm and regulated, it can help handle issues associated with affordability much effectively [36].

 It should be stressed that creating private-public sector partnerships and collaborations with the private sector create the potential to devise effective policy instruments in this regard. Contracting out, licensing, franchising, partnerships etc., are some frequent and viable public-private interventions. Moreover, public and private sector collaborations can bring in synergies, create channels that are mutual, which can strengthen private sector resources and sharing of expertise. More importantly, private public partnerships can help negotiate regulation of pricing policies of private healthcare players. The reason being, typically, private sector health care facilities are considered costly and this keeps many patients away from accessing healthcare services. Nevertheless, this can risk lives of chronic patients for whom receiving continuous medical treatment is crucial. Hence, collaborations can lead towards achieving an effective and affordable service offering that can also enable equitable access for healthcare facilities. By extending licensing and accreditation systems to private healthcare operators, quality of private sector healthcare facilities can be further strengthened. Countries like Brazil, South Africa etc., benefit from successful implementation of such interventions. [67]. 

 Diversifying risks by pooling to a fund for mutual benefit can be proposed as feasible solutions, which can be considered under development of strategies and policies in this regard. This is valid in a context to reduce healthcare expenses associated with persistent diseases such as NCDs, which require continuous treatment. Commercial insurance and community-based mutual services are some practical examples. Those suffering from brain diseases such as, epilepsy, mental retardation and chronic headache as well as cancer and cardiac diseases can immensely benefit from such services. Paving the way for affordable healthcare facilities, developing countries like Colombia, Ghana etc., have implemented such insurance schemes. This can reduce the financial burden, and enhance equitable access to healthcare services."

---

## [Decision Letter · Decision Letter 1]

10 Sep 2020

Chronic Diseases: An Added Burden to Income and Expenses of Chronically-ill People in Sri Lanka

PONE-D-20-08967R1

Dear Dr. Jayathilaka,

We’re pleased to inform you that your manuscript has been judged scientifically suitable for publication and will be formally accepted for publication once it meets all outstanding technical requirements.

Kind regards,

Khurshid Alam, Ph. D.

Academic Editor

PLOS ONE

Additional Editor Comments (optional):

Reviewers' comments:

Reviewer's Responses to Questions

**Comments to the Author**

1. If the authors have adequately addressed your comments raised in a previous round of review and you feel that this manuscript is now acceptable for publication, you may indicate that here to bypass the “Comments to the Author” section, enter your conflict of interest statement in the “Confidential to Editor” section, and submit your "Accept" recommendation.

Reviewer #1: All comments have been addressed

Reviewer #2: All comments have been addressed

2. Is the manuscript technically sound, and do the data support the conclusions?

Reviewer #1: Yes

Reviewer #2: Yes

3. Has the statistical analysis been performed appropriately and rigorously? 

Reviewer #1: Yes

Reviewer #2: Yes

4. Have the authors made all data underlying the findings in their manuscript fully available?

Reviewer #1: No

Reviewer #2: Yes

5. Is the manuscript presented in an intelligible fashion and written in standard English?

Reviewer #1: Yes

Reviewer #2: Yes

6. Review Comments to the Author

Reviewer #1: No comments. Authors have addressed all my comments and I am satisfied with the revised version. All the best!

Reviewer #2: Authors have sufficiently revised the manuscript according to previous review. Therefore, I would like to accept the revised manuscript.

7. PLOS authors have the option to publish the peer review history of their article (what does this mean?). If published, this will include your full peer review and any attached files.

Reviewer #1: No

Reviewer #2: No

---

## [Editor Report · Acceptance letter]

18 Sep 2020

PONE-D-20-08967R1 

Chronic Diseases: An Added Burden to Income and Expenses of Chronically-ill People in Sri Lanka 

Dear Dr. Jayathilaka:

I'm pleased to inform you that your manuscript has been deemed suitable for publication in PLOS ONE. Congratulations! Your manuscript is now with our production department. 

Kind regards, 

on behalf of

Dr. Khurshid Alam 

Academic Editor

PLOS ONE